# Prognostic Value of Prospective Longitudinal CRP to Albumin Ratio among Older Outpatients with Cancer

**DOI:** 10.3390/cancers13225782

**Published:** 2021-11-18

**Authors:** Fiamma Burgassi, Elena Paillaud, Johanne Poisson, Guilhem Bousquet, Frédéric Pamoukdjian

**Affiliations:** 1Service de Médecine Gériatrique, Hôpital Avicenne, APHP, 93000 Bobigny, France; fiamma.burgassi@aphp.fr; 2Service de Gériatrie, Hôpital Européen Georges Pompidou, APHP, 75015 Paris, France; elena.paillaud@aphp.fr (E.P.); johanne.poisson@aphp.fr (J.P.); 3Faculty of Health, University of Paris, 75006 Paris, France; 4Clinical Epidemiology and Ageing, IMRB-UPEC/Inserm U955, Université Paris-Est Créteil, 94000 Créteil, France; 5Service d’Oncologie Médicale, Hôpital Avicenne, APHP, 93000 Bobigny, France; guilhem.bousquet@aphp.fr; 6Inserm UMR_S942, Cardiovascular Markers in Stressed Conditions, MASCOT, Université Sorbonne Paris Nord, 93000 Bobigny, France

**Keywords:** cancer-cachexia, longitudinal trajectories, CRP to albumin ratio, older adults, geriatric assessment

## Abstract

**Simple Summary:**

The prognostic value of the C-reactive protein to albumin ratio (CAR) among older adults with cancer is not known. As an inflammation and nutrition-based score, the CAR could be used as a biomarker of cancer-cachexia. We aimed to assess the prognostic value of longitudinal trajectories of the CAR on overall survival among older adults with cancer. By identifying two distinct clusters in the longitudinal trajectories of the CAR with significantly different overall survivals, we were able to characterize older patients with cancer which are the most at-risk to have a cancer-cachexia trajectory. For these patients (typically the most frail with a metastatic cancer), we suggest an early assessment of muscle mass in order to start a multimodal rehabilitation as soon as possible.

**Abstract:**

The prognostic value of the CRP to albumin ratio (CAR) among older adults with cancer is not known. Six hundred and three older outpatients with cancer and undergoing geriatric assessment before therapeutic decisions were prospectively recruited from the PF-EC cohort study. Serum albumin levels, serum CRP levels and the CAR were prospectively recorded at baseline, and at each consultation thereafter, as follows: 1, 3, 6, 9, 12, 18, 24 and 36 months. Frailty was defined as a G8-index ≤ 14. The primary endpoint was longitudinal variation in the CAR during the study follow-up. Two clusters in the longitudinal trajectories of the CAR were identified, one favourable, with lower values and better overall survival (cluster A), and the second with higher values and less favourable overall survival (cluster B). The median CAR [95% CI] for clusters A and B were respectively: 0.17 [0.04–0.48] and 0.26 [0.04–0.79] at baseline (*p* = 0.01), and 0.18 [0.02–3.17] and 0.76 [0.03–6.87] during the study follow-up (*p* < 0.0001). Cluster B was associated with the frailest patients with metastatic disease, mainly driven by a high CRP level at baseline, and low albumin during the study follow-up. Our study results suggest that the most risk-prone patients have a cancer-cachexia trajectory.

## 1. Introduction

Cancer-cachexia (CC) is a wasting syndrome characterized by weight loss with concomitant loss of muscle and/or fat mass, which cannot be fully reversed by conventional nutritional support and which leads to progressive functional impairment [1]. CC results in a negative protein and energy balance (malnutrition) caused by reduced food intake and excessive catabolism (inflammation) [2].

As a host response to cancer-mediated by cytokines, systemic inflammation often occurs with an activation of the hepatic acute-phase protein response, which can lead to increased levels of C-reactive protein (CRP) and decreased levels of albumin [2,3]. The modified Glasgow score, serving as an inflammation and nutrition-based prognostic score, which combines CRP and albumin levels (CRP ≤ 10 mg/L + albumin ≥ 35 g/L; CRP ≤ 10 mg/L + albumin < 35 g/L; CRP > 10 mg/L or CRP > 10 mg/L + albumin < 35 g/L), has been shown to be significantly associated with overall survival, independently from cancer extension or cancer treatment [4]. More recently, the CRP/albumin ratio (CAR) has been proposed as a new prognostic index combining inflammation and nutrition approaches [5]. In a recent meta-analysis including twenty-seven studies and a total of 10,556 cases with various solid tumours, a high CAR was associated with shorter survival [6]. In addition, few observational studies have shown the superiority of the CAR in predicting overall survival over other prognostic scores combining inflammation and/or nutritional parameters, notably the neutrophil to lymphocyte ratio (NLR), the platelet to lymphocyte ratio (PLR), and the Glasgow prognostic score (GPS). Indeed, by comparison with the NLR, the PLR, and the GPS, the CAR was found to have a better discrimination (AUC) in predicting overall mortality in gastric and oesophageal cancers [7,8].

To date, the prognostic value of the CAR among older adults with cancer is not known.

We aimed to assess the prognostic value of longitudinal trajectories of the CAR in a prospective cohort study of older outpatients with cancer.

## 2. Materials and Methods

### 2.1. Study Design and Patients

Patients were recruited from the Physical Frailty in Elderly Cancer patients (PF-EC) study. This study was a prospective, observational, two-centre cohort study that was initiated in November 2013 and has already been described [9]. In brief, all consecutive older in- and outpatients with cancer, aged 65 and over who were referred for a geriatric assessment (GA) before any cancer treatment decision were included on 30 September 2017.

The inclusion date was considered to be the date of the patient’s first geriatric-oncology consultation.

All patients provided their informed consent before inclusion in the study. The study was approved by the local independent ethics committee (Avicenne Hospital, Bobigny, France; reference: CLEA-2015-019).

### 2.2. Cancer-Related and Demographic Data

Demographic data (age and sex), cancer-related data (cancer site and extension: local or metastatic) and the Eastern Cooperative Oncology Group Performance Status (ECOG-PS) were recorded at the first geriatric oncology consultation, as part of the geriatric assessment. The type of treatment (surgery, chemotherapy, radiotherapy, hormone therapy, targeted therapy, percutaneous resection or intra-arterial treatment for liver cancer, or exclusively supportive care) received by each patient was recorded at the first six-month follow-up consultation.

### 2.3. Geriatric Assessment (GA)

The GA was conducted during the patient’s first geriatric oncology consultation. The GA involved eight domains as follows: social environment (self-reported question: do you live at home on your own? Yes/No); comorbidities (total Cumulative Illness Rating Scale for Geriatrics (CIRS(G)) considered as abnormal above the median value of 14) [10]; poly-medication (≥5 drugs a day) [11]; dependency (activities of daily living (ADL) score ≤ 5/6, and/or instrumental ADL (IADL) score ≤ 3/4) [12,13]; malnutrition (body mass index (BMI) < 21 kg/m^2^ [14]; impaired mobility (gait speed < 0.8 m/s) [9]; depressed mood (Mini-Geriatric Depression Scale (Mini-GDS) score ≥ 1/4) [15]; and cognitive impairment (Mini-Mental State Examination (MMSE) score < 24/30) [16].

### 2.4. Definition of Frailty

Frailty was defined as an abnormal G8-index ≤ 14/17 [17]. We also considered the G8-index as a continuous variable.

### 2.5. Measurement of the CRP/Albumin Ratio (CAR)

The serum albumin level (g/L), the serum C-reactive protein (CRP, mg/L) level and the CAR were prospectively recorded at each consultation. For our purposes, these variables were considered as continuous.

### 2.6. Outcomes

Follow-up visits during the first 36 months were planned as follows: at 1, 3, 6, 9, 12, 18, 24 and 36 months respectively.

The main outcome was variation in the CRP/albumin ratio (CAR) during study follow-up.

The secondary outcome was long-term mortality. Vital status was determined by telephoning the patients or their family or by extracting data from medical records.

### 2.7. Statistical Analyses

The data were analysed using R statistical software (version 4.0.0; R Foundation for Statistical Computing, Vienna, Austria).

Descriptive analysis: categorical variables were summarized as numbers (percentages), and continuous variables were summarized as means ± standard deviation (SD) or medians ± interquartile range (IQR) as appropriate.

Clustering of longitudinal trajectories of the CAR in the first 36 months of follow-up: to explore the existence of homogeneous patient CAR trajectories, we used k-means as a partitioning method. K-means is an algorithm based on expectation (E) and maximization (M) after assigning each patient to a cluster. Alternation between E and M is repeated several times until the optimal partition is found. Here, we used the k-mean longitudinal (KML) package in R, which runs k-means between two and six clusters, 20 times each. To choose the optimal number of clusters based on the quality of partition, we used the Calinski and Harabatz criterion, one of the most popular indices for assessing longitudinal data [18]. The Calinski and Harabatz index is the intergroup/intragroup variance ratio which helps to maximize the distance between clusters. A multivariate analysis of variance (MANOVA) was then carried out to assess whether longitudinal cluster trajectories were significantly different according to clusters and measurement times. The mean change in CAR, CRP and albumin levels according to clusters are graphically presented in a line chart.

Phenotypes associated with CRP/Albumin ratio clusters: a comparison between clusters was made with the data collected using Student’s *t* test or Wilcoxon’s test for quantitative variables, and chi square test or Wald’s test for categorical variables, as appropriate. Multivariate logistic regression was performed to assess the strength of associations between variables and clusters, expressed as an adjusted odds ratio (aOR; 95% confidence interval [CI]). Variables yielding p values under 0.20 in univariate analysis were considered for inclusion in the multivariate analysis. A backward selection process of the highest *p* values was performed to retain the final model. The interaction between multivariate factors was also checked.

Survival analysis: median survival and survival curves according to clusters A/B were determined using the Kaplan–Meier method. A univariate Cox proportional hazard regression model was run to assess the association between clusters A/B and overall survival, expressed as a hazard ratio (HR; 95% confidence interval [CI]). We also assessed the association between the baseline CAR as continuous variable and overall survival. The *p* value was determined using the log-rank test.

All the tests were two-sided, and the threshold for statistical significance was set at a *p* value of less than 5%. Multivariate imputation by chained equations was used to handle missing data at baseline for albumin (n = 100) and CRP (n = 112).

## 3. Results

### 3.1. Patients

Of the 959 consecutive elderly patients with cancer in the PF-EC cohort who had been referred for a GA up to 30 September 2017, 356 were excluded because they were inpatients and were not followed up over time. Hence, 603 outpatients were included in the present study.

### 3.2. Baseline Characteristics of Patients and the G8 Frailty Index

The mean age ± SD of the study population was 81.2 ± 6.1 years. Most of the patients were women (54%), with solid tumours (94%) and local cancer (55%). Colorectal and breast cancers were the two most common types. Eighty-nine % (n = 535/603) of the patients were frail according to a G8 index score ≤ 14. In the GA, the different geriatric domains were variably impaired, from 14% for BMI to 67% for polymedication (Table 1).

At the time of the first GA, the median CAR was 0.20 [0.04–0.62].

### 3.3. Clustering Longitudinal CAR Trajectories

The median follow-up time was 15.3 months [6.0–31.0] (min-max: 0–66). With a mean number of 6.0 ± 2.4 measures of the CAR for each patient, CAR values ranged from 0 to 35.8, and the median CAR was 0.25 [0.03–4.96].

Two longitudinal CAR clusters (A and B) were identified as the optimal partition (Figure 1a). Overall, the median CARs for clusters A and B were respectively: 0.17 [0.04–0.48] and 0.26 [0.04–0.79] (*p* = 0.01) at baseline, and 0.18 [0.02–3.17] and 0.76 [0.03–6.87] (*p* < 0.0001) during the study follow-up. The profile of cluster A was more stable over time, with significantly lower CAR values than for cluster B. Cluster B was associated with significantly higher CAR values than cluster A, with two peaks occurring later at 9 and 24 months (Table A1).

Compared to baseline, CAR peaks were correlated with the highest CRP values and the lowest albumin values (Figure 1b,c). Individual trajectories for each cluster are shown in Figure A1. Considered individually, the variations of CRP and albumin were significantly different (*p* value for MANOVA < 0.0001) according to the two clusters during study follow-up.

In stratified analysis by cancer site, we found that the longitudinal profiles of the two clusters remained similar in breast, colorectal and lung cancers respectively (Figure A2)

Thus, two longitudinal CAR clusters were identified, one favourable with lower values (cluster A) and the second with a less favourable profile (cluster B).

### 3.4. Baseline Phenotype Associated with CAR Cluster B

In univariate analysis, the G8 frailty index as a continuous variable, the cancer site, metastatic status, cancer treatment modalities (mainly surgery, hormone therapy, exclusively supportive care), the social environment (not living alone) and ADL-dependency (≤5/6) were significantly associated with CAR cluster B (Table 2). Further analysis showed a reverse linear association between the G8 frailty index and CAR cluster B (Figure 2).

In multivariate analysis, the G8 frailty index and the social environment were negatively associated with CAR cluster B, while the cancer site (mainly digestive cancers) and metastatic status were positively associated with CAR cluster B (Table 2). There was no significant interaction between living alone and the cancer site (*p* = 0.32), and no interaction between living alone and metastatic status (*p* = 0.84).

### 3.5. Long-Term Suvival and CAR Clusters

Over the study follow-up, the mortality rate was 47% (n = 282/603). Median overall survival was 32.0 months [27–37].

According to CAR clusters, the median survival time was 37.0 months [31.0–48.5] and 25.0 months [18.5–34.0] for clusters A and B respectively (Figure 3). Compared to cluster A, overall survival for cluster B was significantly lower with a HR = 1.39 [1.09–1.75], *p* = 0.007. The baseline CAR values per 1.5 ranges of more was also significantly associated with overall survival with similar predictive values: HR = 1.23 [1.17–1.29], *p* < 0.0001.

According to the baseline CRP, the median survival time was 47.0 months [37.0-NA] and 12.5 months [8.90–19.6] for CRP ≤ 10 mg/L (n = 352) and CRP > 10 mg/L (n = 251) respectively.

According to the baseline albumin, the median survival time was 39.7 months [34.5–48.3] and 13.5 months [9.30–21.2] for albumin ≥ 35 g/L (n = 419) and albumin < 35 g/L (n = 184) respectively.

As continuous variable at baseline, the HRs [95% CI] were 1.24 [1.18–1.30], 1.01 [1.00–1.01] and 1.06 [1.05–1.08] for CAR (per 1.5 *SD* more), CRP (per 35.0 *SD* more) and albumin (per 6.5 *SD* less) respectively. By reference for the CAR at baseline, the HRs at baseline for CRP and albumin were significantly lower (*p* value for head-to-head comparison of HRs < 0.0001).

## 4. Discussion

In this study two baseline clusters for longitudinal CAR trajectories were identified, one favourable with lower values and better overall survival (cluster A), and the second with higher values and less favourable overall survival (cluster B). In particular, in our prospective cohort of 603 older patients with cancer, the initial CAR value was an accurate predictive biomarker of survival.

This important result has potential clinical applications, particularly for patients belonging to the unfavourable cluster B. Cluster B was associated with the frailest patients with larger tumour mass (i.e., metastatic disease) mainly driven by high inflammation (CRP) at baseline and a negative protein balance (albumin) during study follow-up, suggesting a more aggressive cancer with a high risk of cancer-cachexia [19]. Thus, cluster B could also capture progression of cancer disease, particularly in metastatic setting, which may explain the impressive peak at 9 months. Indeed, the median progression free survival in the first line setting for metastatic cancer is ranging from 5 months to 10 months in lung, breast and colorectal cancers [20,21,22]. In addition, an overlap pathway between cancer-cachexia and frailty syndrome in the same cohort has previously been shown [23]. While we did not assess muscle mass in this study, our study results suggest that the baseline phenotype of CAR cluster B is more likely to belong to a cancer-cachexia trajectory [24]. As countering tumour mass remains the most effective treatment against inflammation, optimal cancer treatment could reduce the likelihood of a cancer-cachexia trajectory for these patients. Recently, we published a simple clinical score (namely the GRADE) based on weight loss, gait speed, cancer site and extension, to help in cancer-treatment decisions and to limit situations of over- and undertreatment among older patients with cancer [25]. A GRADE score < 11 suggests a favourable cost/benefit ratio to treat cancer optimally, which could be particularly relevant for patients belonging to CAR cluster B.

Furthermore, the longitudinal profile of cluster B showed higher levels of the CAR occurring later during study follow-up. This study result could be explained by the decline in immune function with age (also called immune-senescence), which includes inflamm-aging especially for the frailest elderly subjects [26]. Typically, inflamm-aging is associated with increased levels of proinflammatory cytokines and reduced levels of anti-inflammatory cytokines. Common pathways of cancer-cachexia are summarized in Figure A3 (adapted from [3,19]). From a clinical point of view, in the context of a cytokine storm involving metastatic cancer and frailty among older adults (cluster B in particular), we suggest an early assessment of muscle mass in order to start a multimodal approach as soon as possible, which would lead to a reduction in the tumour-associated inflammation and an increase in anabolism and appetite [24]: muscle, functional and nutritional rehabilitation with a complementary pharmacological approach using megestrol acetate, possibly combined with short-term high-dose corticosteroid-therapy.

Further studies are required to assess (i) the longitudinal phenotype associated with cluster B; (ii) to compare the longitudinal CAR between younger and older adults with cancer, and iii) to assess other well-known biomarker leading to cancer-cachexia, such as proinflammatory cytokines (e.g., Tumour Necrosis Factor-α or Interleukin-6) among older cancer patients.

## 5. Conclusions

We have identified two clusters of longitudinal trajectories for the CAR with significantly different overall survival rates among older adults with cancer. The worst profile was mainly underpinned by cancer-related inflammation at baseline, followed by a negative protein balance during study follow-up. Our study results suggest that these patients are the most risk-prone patients for a cancer-cachexia trajectory for which an early assessment of muscle mass is suggested.

## Figures and Tables

**Figure 1 cancers-13-05782-f001:**
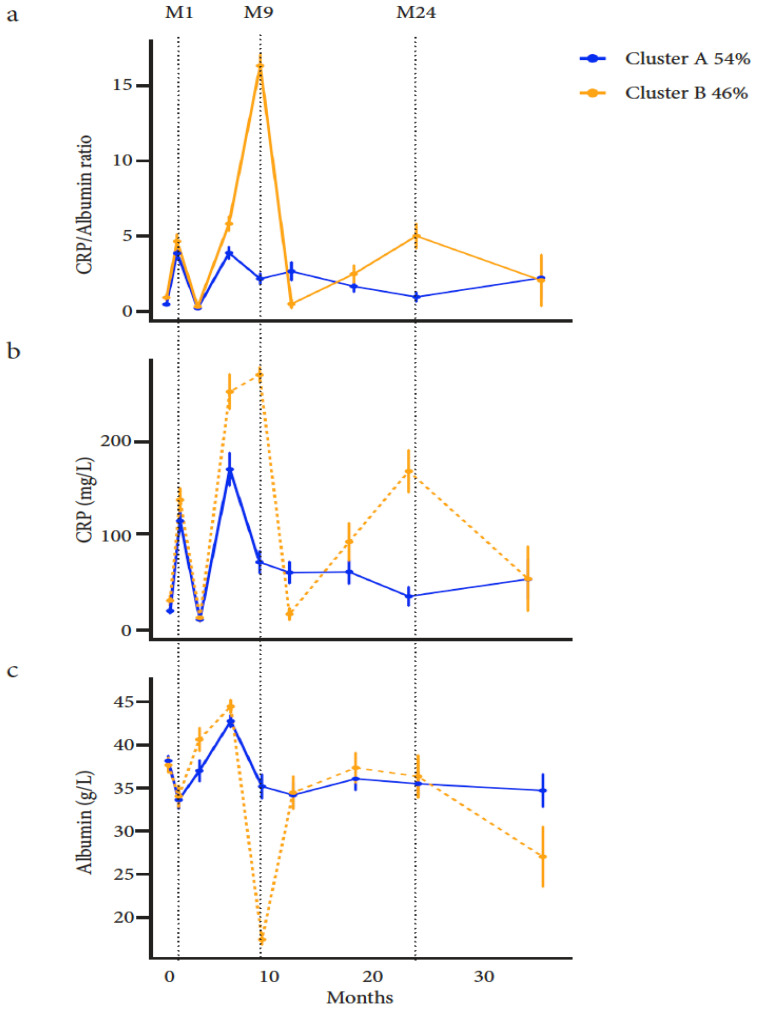
(**a**): Longitudinal trajectory clustering of the CAR among elderly outpatients with cancer; (**b**): longitudinal trajectory of CRP levels among elderly outpatients with cancer; (**c**): longitudinal trajectory of albumin levels among elderly outpatients with cancer.

**Figure 2 cancers-13-05782-f002:**
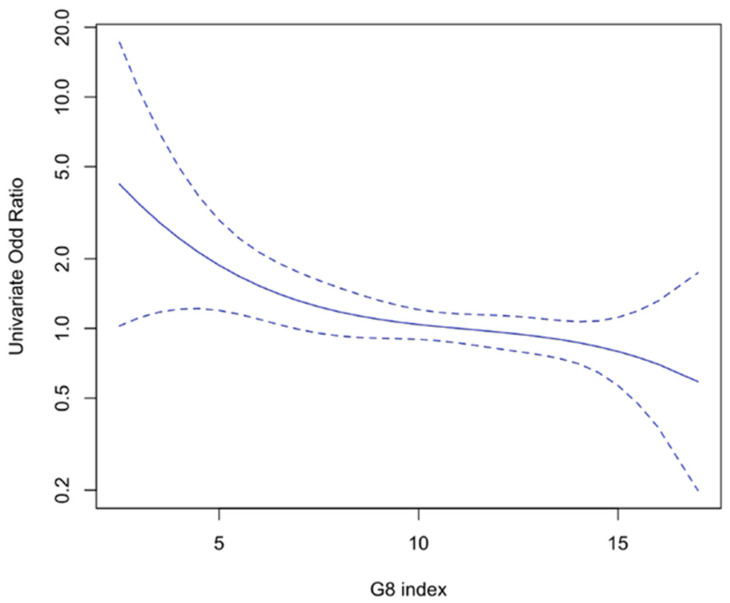
Linear association between the G8 index and CAR cluster B among elderly cancer patients.

**Figure 3 cancers-13-05782-f003:**
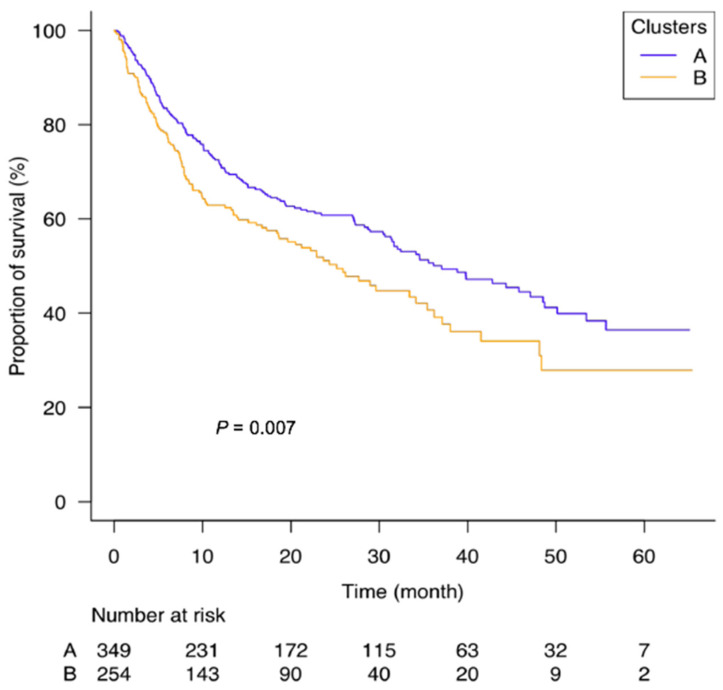
Kaplan–Meier survival curves for long-term mortality among elderly cancer patients in relation to CAR clusters.

**Table 1 cancers-13-05782-t001:** Baseline characteristics of the 603 older outpatients with cancer in the PF-EC cohort.

Variables	Whole Cohort N (%)
Age (y), mean ± SD	81.2 ± 6.1
Sex ratio (male/female)	280 (46)/323 (54)
G8 index, median ± IQR	11.0 ± 4.0
Cancer site: Colorectal Breast Lung Liver Digestive non-colorectal Genital or urinary tract Haematological malignancies Skin, melanoma Prostate Other *	109 (18) 105 (17) 92 (15) 85 (14) 80 (13) 40 (7) 34 (6) 16 (3) 16 (3) 26 (4)
Metastatic (yes)	271 (45)
Cancer-treatment modalities: Surgery Chemotherapy Radiotherapy Hormone therapy Targeted therapy Percutaneous resection (liver cancer) Intra-arterial treatment (liver cancer) Exclusively supportive care	172 (28.5) 200 (33) 131 (22) 96 (16) 43 (7) 41 (7) 22 (4) 124 (20.5)
ECOG-PS > 2	204 (34)
Living alone (yes)	241 (40)
Comorbidities CIRSG total > 14	269 (45)
Polymedication (yes)	402 (67)
Dependency ADL ≤ 5/6 IADL ≤ 3/4	204 (34) 386 (64)
Nutrition BMI < 21 kg/m^2^ CRP (mg/L), median [Q1–Q3] Albumin (g/L), median [Q1–Q3]	82 (14) 7.1 [1.8–21.5] 38.0 [33.6–42.0]
Mobility Gait speed < 0.8 m/s Gait speed (m/s), median ± IQR	345 (57.5) 0.73 ± 0.48
Mood (n = 597) Mini-GDS ≥ 1/4	261 (44)
Cognition (n = 429) MMSE < 24/30	217 (51)

IQR = Interquartile range. * Sarcoma (n = 5), mesothelioma (n = 8), unknown primary site (n = 10), head and neck (n = 3), thymus (n = 1) ECOG-PS: Eastern Cooperative Oncology Group Performance Status; CIRSG: Cumulative Illness Rating Scale Geriatric; ADL: activities of daily living; IADL: instrumental-ADL; BMI: body mass index; CRP: C-reactive protein; Mini GDS: Mini Geriatric Depression Scale; MMSE: Mini Mental State Examination.

**Table 2 cancers-13-05782-t002:** Univariate and multivariate factors associated with CAR cluster B.

Variables	Cluster A N = 349 (%)	Cluster B N = 254 (%)	*p* *	Adjusted Odd Ratio [95% CI]	*p **
Age (y), mean ± SD	80.9 ± 6.1	81.6 ± 6.2	0.20		
Sex ratio (male/female)	164 (47)/185 (53)	116 (46)/138 (54)	0.75		
Frailty-G8 index (≤14) G8 index, median ± IQR	303 (87) 11.5 ± 4.0	232 (91) 11.0 ± 4.5	0.08 0.01	0.94 [0.88–0.99]	0.04
Cancer site: Breast Colorectal Lung Liver Digestive non-colorectal Genital or urinary tract Haematological malignancies Skin, melanoma Prostate Other **	70 (20) 74 (21) 51 (15) 44 (13) 38 (11) 19 (5) 17(5) 6 (1.5) 14 (4) 16 (4.5)	35 (14) 35 (14) 41 (16) 41 (16) 42 (16) 21 (8) 17 (7) 10 (4) 2 (1) 10 (4)	0.007	1 (reference) 0.80 [0.44–1.43] 1.15 [0.63–2.12] 1.57 [0.85–2.86] 1.87 [1.01–3.45] 1.57 [0.73–3.40] 1.43 [0.63–3.22] 2.59 [0.84–7.98] 0.17 [0.03–0.85] 0.90 [0.36–2.27]	0.02
Metastatic (yes)	138 (39.5)	133 (52)	0.001	1.80 [1.27–2.55]	0.001
Cancer-treatment modalities: Surgery Chemotherapy Radiotherapy Hormone therapy Targeted therapy Percutaneous resection (liver cancer) Intra-arterial treatment (liver cancer) Exclusively supportive care	114 (33) 122 (35) 84 (24) 67 (19) 27 (8) 25 (7) 9 (2.5) 59 (17)	58 (23) 78 (31) 47 (18.5) 29 (11) 16 (6) 16 (6) 13 (5) 65 (25.6)	0.008 0.27 0.10 0.009 0.50 0.67 0.10 0.009	- - - - -	
ECOG-PS > 2	107 (31)	97 (38)	0.05	-	
Living alone (yes)	154 (44)	87 (34)	0.01	0.69 [0.48–0.98]	0.03
Comorbidities CIRSG total > 14	151 (43)	118 (46)	0.44		
Polymedication (yes)	238 (68)	164 (64.5)	0.35		
Dependency ADL ≤ 5/6 IADL ≤ 3/4	106 (30) 213 (61)	98 (38.5) 173 (68)	0.03 0.07	- -	
Nutrition BMI < 21 kg/m^2^	44 (13)	41 (16)	0.21		
Mobility Gait speed < 0.8 m/s Gait speed (m/s), median ± IQR	200 (57) 0.75 ± 0.46	145 (57) 0.70 ± 0.50	0.91 0.22		
Mood (n = 597) Mini-GDS ≥ 1/4	152 (43.5)	109 (43)	0.96		
Cognition (n = 429) MMSE < 24/30	131 (37.5)	86 (34)	0.83		

* Chi squared test or Student’s *t* test or Wilcoxon’s test as appropriate; bold = significant *p* value at the threshold of 5%. ** Sarcoma (n = 5), mesothelioma (n = 8), unknown primary site (n = 10), head and neck (n = 3), thymus (n = 1). ECOG-PS: Eastern Cooperative Oncology Group Performance Status; CIRSG: Cumulative Illness Rating Scale Geriatric; ADL: activities of daily living; IADL: instrumental-ADL; BMI: body mass index; CRP: C-reactive protein; Mini GDS: Mini Geriatric Depression Scale; MMSE: Mini Mental State Examination.

## Data Availability

The data presented in this study are available on request from the corresponding author.

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
