# Peer review of "Prognostic Value of Prospective Longitudinal CRP to Albumin Ratio among Older Outpatients with Cancer"

_cancers, 2021, doi:10.3390/cancers13225782_

Round 1

Reviewer 1 Report

To the Authors,
please find below my evaluations and suggestions regarding the manuscript: Prognostic value of prospective longitudinal CRP to Albumin ratio among older outpatients with cancer.

The manuscript is really easy to follow and well written; Authors avoided any excessive explanations and kept their work fluent. Data are convincing and technically correct; the statistic part is really well explained and performed. Those reported below are curiosities rather than concerns.

- When Authors report the “superiority” of the CAR compared to the other index in the Introduction section, they should add a brief sentence in which they explain how this index is more powerful compared to the others.

- As a personal preference, I would move the graphs of individual trajectories from Appendix B to Fig.1, this would help the understanding of the data and would give more compactness to the manuscript, but as I said, this is not mandatory.

- It would be possible to extrapolate subclusters from cluster A and B? Specific cancer-related subclusters (and their CARs) would be really intriguing.

- Which is the Authors’ explanation for the impressive peak for Cluster B at M9?

- Considering the longitudinal profile of cluster B and the higher levels of CARs during study follow-up, were the people belonging to that cluster under medication at the moment of consultation? In addition to what Authors proposed (the decline in immune response and immune-aging), would be possible to speculate that the higher levels of CAR in cluster B were also “caused” by a “not completely adequate” medication proposed following a “not completely adequate” index?

English is fine; no typing mistakes detected.

Author Response

Reviewer 1

The manuscript is really easy to follow and well written; Authors avoided any excessive explanations and kept their work fluent. Data are convincing and technically correct; the statistic part is really well explained and performed. Those reported below are curiosities rather than concerns.

We would like to thank Reviewer 1 for highlighting the quality of writing and the methodology of our study.

- When Authors report the “superiority” of the CAR compared to the other index in the Introduction section, they should add a brief sentence in which they explain how this index is more powerful compared to the others.

To follow this remark by Reviewer 1, we have now added the following sentence to explain how the CAR is considered by authors from 2 observational studies to be superior to the others index:

Line 62 “Indeed, by comparison with the Neutrophil to Lymphocyte Ratio, the Platelet to Lymphocyte Ratio, and the Glasgow Prognostic Score, the CAR was found to have a better discrimination (AUC) in predicting overall mortality in gastric and esophageal cancers [7, 8]”.

- As a personal preference, I would move the graphs of individual trajectories from Appendix B to Fig.1, this would help the understanding of the data and would give more compactness to the manuscript, but as I said, this is not mandatory.

We thank Reviewer 1 for this proposition; however, we do not think it brings too much to Figure 1 as it was in the initial version submitted for revision.

- It would be possible to extrapolate subclusters from cluster A and B? Specific cancer-related subclusters (and their CARs) would be really intriguing.

Thank you for your interesting comment. To follow the remark by Reviewer 1, we have now added a supplemental Appendix Figure B to show the two clusters A/B in stratified analysis by cancer site: Breast, Colorectal, and Lung. Overall, the longitudinal profiles of the two clusters remained similar to the whole analysis.

We have added this information in results chapter as follows: Line 181“In stratified analysis by cancer site, we found that the longitudinal profiles of the two clusters remained similar in breast, colorectal and lung cancers respectively (Figure B2).”

- Which is the Authors’ explanation for the impressive peak for Cluster B at M9?

By combining inflammation and nutrition parameters, the CAR could be a simple biomarker of cancer cachexia trajectory. We thus believe that this peak of CAR value at 9-month is related to the occurrence of cancer cachexia, itself driven by inflammation (high CRP) first and then malnutrition (low albumin). A hypothesis to explain the peak at 9-month could be a progression of cancer disease. In our cohort, we did not collect data for progression free survival. However, considering that Cluster B is significantly associated with metastatic disease, we think that it is closely linked to the progression of cancer disease, particularly in metastatic setting. Indeed, in the literature, in the first line setting for metastatic cancer, median PFS is ranging from 5 months to 10 months in lung, breast and colorectal cancers (Rossi A et al. Lancet Oncol, 2014; Kiely BE et al. J Clin Oncol, 2011; Dagher M et al. J Cancer Res Clin Oncol, 2019).

We have now added this information in discussion as follows: Line 237 “Cluster B was associated with the frailest patients with larger tumor mass (i.e. metastatic disease) mainly driven by high inflammation (CRP) at baseline and a negative protein balance (albumin) during study follow-up, suggesting a more aggressive cancer with a high risk of cancer-cachexia [19].   Thus, cluster B could also capture progression of cancer disease, particularly in metastatic setting, which may explain the impressive peak at 9-month. Indeed, the median progression free survival in the first line setting for metastatic cancer is ranging from 5 months to 10 months in lung, breast and colorectal cancers (Rossi A et al. Lancet Oncol, 2014; Kiely BE et al. J Clin Oncol, 2011; Dagher M et al. J Cancer Res Clin Oncol, 2019)

- Considering the longitudinal profile of cluster B and the higher levels of CARs during study follow-up, were the people belonging to that cluster under medication at the moment of consultation?

All patients included in this study were assessed before any cancer treatment decision and thus were not under specific medication at the time of inclusion.

We have now added this information in the “methods” chapter as follows: Line 73 “In brief, all consecutive older in- and outpatients with cancer, aged 65 and over who were referred for a Geriatric Assessment (GA) before a cancer treatment decision were included on 30 September 2017.

In addition to what Authors proposed (the decline in immune response and immune-aging), would be possible to speculate that the higher levels of CAR in cluster B were also “caused” by a “not completely adequate” medication proposed following a “not completely adequate” index?

All patients included in this study undergone a Geriatric Assessment before any cancer treatment decision. Thus, this is a post-hoc analysis based on prospectively collected data. CAR index was not used initially to help cancer treatment decision, the latter being based on a weekly multidisciplinary consultation meeting.

English is fine; no typing mistakes detected.

Reviewer 2 Report

This is a small prospective study and the prognostic value of the CRP to Albumin Ratio (CAR) among older adults with cancer was investigated.

I have several significant concerns listed below. Manuscript can be improved addressing these comments.

I am not quite sure how these two clusters were identified. Please elaborate.

Simple summary and abstract seem to be inconsistent. I would suggest to be consistent.

There is no control such as younger patients to compare.

Line 18: spell out CRP

Line 27: Spell out  603

Table 1: female number should be listed

Figure 1a: CRP is not very different between A and B. Some reason Albumin levels were very low at 10 months in Cluster B. CRP/ Albumin ratio is very different at 10 months, but it is mainly due to difference in Albumin levels. Please explain how Albumin levels is so low in Cluster B at 10 months. If the Albumin level is deciding factor, please justify the benefit using CAR instead of Albumin level in this study.

Author Response

Reviewer 2

This is a small prospective study and the prognostic value of the CRP to Albumin Ratio (CAR) among older adults with cancer was investigated.

Most of observational studies in elderly cancer patients using geriatric assessment usually have between 37 and 650 patients (see Handforth C et al. The prevalence and outcomes of frailty in older cancer patients: A systematic review. Ann Oncol, 2015).

Our study, which has 603 patients can be considered as large.

I have several significant concerns listed below. Manuscript can be improved addressing these comments.

I am not quite sure how these two clusters were identified. Please elaborate.

In the initial version of our manuscript, we have retained: “to explore the existence of homogeneous patient CAR trajectories we used k-means as a partitioning method. K-means is an algorithm based on expectation (E) and maximization (M) after assigning each patient to a cluster. Alternation between E and M is repeated several times until the optimal partition is found. Here, we used the K Mean Longitudinal (KML) package in R, which runs k-means between 2 and 6 clusters, 20 times each. To choose the optimal number of clusters based on the quality of partition, we used the Calinski and Harabazt criterion, one of the most popular indices for assessing longitudinal data [18]. A multi-variate analysis of variance (MANOVA) was then carried out to assess whether longitudinal cluster trajectories were significantly different according to clusters and measurement times. The mean change in CAR, CRP and albumin levels according to clusters are graphically presented in a line chart.”

To better explain how the two clusters were identified, we brought the following precision regarding the Calinski Harabast criteria : Line 132 “The Calinski and Harabazt indice is the inter-group / intra-group variance ratio which help to maximize the distance between clusters.”

Simple summary and abstract seem to be inconsistent. I would suggest to be consistent.

We do not understand this remark by Reviewer 2, since the simple summary, restricted to less than 150 words according to the Cancers guidelines, is itself an abstract of the abstract. 

There is no control such as younger patients to compare.

Our study was performed using the prospective PF-EC cohort dedicated to elderly with cancers.  However, Reviewer 2 is right since cachexia is a common issue in the progression of cancer whichever the age of the patient.

To our knowledge, such data does not exist in younger patients. Further studies are thus required to see whether the same clusters would be identified in these two populations of age.  

To take into account this remark by Reviewer 2 we added the following sentence in the last part of discussion: Line 262 “Further studies are required to assess i) the longitudinal phenotype associated with cluster B; ii) to compare the longitudinal CAR between younger and older adults with cancer, and iii) to assess other well-known biomarker leading to cancer-cachexia, such as pro-inflammatory cytokines (e.g., Tumour Necrosis Factor-a or Interleukin-6) among older cancer patients.

Line 18: spell out CRP

In agreement with the Reviewer 2, we corrected the sentence as follows: “C-Reactive Protein” line 18

Line 27: Spell out 603

In agreement with the Reviewer 2, we corrected the sentence as follows: “Six hundred and three”

Table 1: female number should be listed

In agreement with the Reviewer 2, we have now added female number (%) in the Table 1 and Table 2.

Figure 1a: CRP is not very different between A and B. Some reason Albumin levels were very low at 10 months in Cluster B. CRP/ Albumin ratio is very different at 10 months, but it is mainly due to difference in Albumin levels. Please explain how Albumin levels is so low in Cluster B at 10 months. If the Albumin level is deciding factor, please justify the benefit using CAR instead of Albumin level in this study.

Thank you for your very interesting comment.

The variation of the CRP during the study follow up was significantly different according to the clusters A and B.

In our study, at baseline, we found that predictive performances of the CAR were significantly superior to CRP or albumin levels.

We have now, added these informations in our manuscript as follows:

  • Line 177 “Considered individually, the variations of CRP and albumin were significantly different (P value for MANOVA < 0.0001) according to the two clusters during study follow-up.”
  • Line 225 “As continuous variable at baseline, the HRs [95%CI] were 1.24 [1.18-1.30], 1.01 [1.00-1.01] and 1.06 [1.05-1.08] for CAR (per 1.5 SD more), CRP (per 35.0 SD more) and albumin (per 6.5 SD less) respectively. By reference for the CAR at baseline, the HRs at baseline for CRP and albumin were significantly lower (P value for head-to-head comparison of HRs < 0.0001)”.

By combining inflammation and nutrition parameters, we were able to capture two significant longitudinal trajectories. Here, the joint reading of the two parameters (see Figure 1a) shows that malnutrition (low albumin levels) follows inflammation (high CRP levels), particularly at 9-month. But this is the CAR value at baseline which is a predictive biomarker of survival and which indicates the need to start an early assessment of muscle mass in order to start a multimodal approach as soon as possible.

Round 2

Reviewer 2 Report

Thank you for the response. I don't have any more comments.